# Activating Self-Attention
# for Multi-Scene Absolute Pose Regression

**Miso Lee**
Sungkyunkwan University
dlalth557@skku.edu

**Jihwan Kim**
Sungkyunkwan University
damien@skku.edu

**Jae-Pil Heo**[†]
Sungkyunkwan University
jaepilheo@skku.edu

## Abstract

Multi-scene absolute pose regression addresses the demand for fast and memory-efficient camera pose estimation across various real-world environments. Nowadays, transformer-based model has been devised to regress the camera pose directly in multi-scenes. Despite its potential, transformer encoders are underutilized due to the collapsed self-attention map, having low representation capacity. This work highlights the problem and investigates it from a new perspective: distortion of query-key embedding space. Based on the statistical analysis, we reveal that queries and keys are mapped in completely different spaces while only a few keys are blended into the query region. This leads to the collapse of the self-attention map as all queries are considered similar to those few keys. Therefore, we propose simple but effective solutions to activate self-attention. Concretely, we present an auxiliary loss that aligns queries and keys, preventing the distortion of query-key space and encouraging the model to find global relations by self-attention. In addition, the fixed sinusoidal positional encoding is adopted instead of undertrained learnable one to reflect appropriate positional clues into the inputs of self-attention. As a result, our approach resolves the aforementioned problem effectively, thus outperforming existing methods in both outdoor and indoor scenes.

## 1 Introduction

Camera pose estimation is a fundamental and essential computer vision task, adopted in numerous applications such as augmented reality and autonomous driving. Geometric pipelines [1–6] with 2D and 3D data have been a mainstream with high accuracy. After extracting features and matching 2D-3D correspondences, camera pose is approximated via Perspective-n-Points (PnP) algorithm and RANSAC [7]. However, there still remains several challenges for real-world applications, including high computational cost and a huge amount of 3D point cloud.

Absolute Pose Regression (APR) tackles these issues by directly estimating the 6-DoF pose from a single RGB image in an end-to-end manner. First introduced by Kendall *et al.* [8], subsequent APR methods [9–18] have been devised based on convolutional neural networks (CNN). However, they still demand multiple models and individual optimization to be applied in real-world multi-scene scenarios. In this regard, Multi-Scene Absolute Pose Regression (MS-APR) has emerged to satisfy the needs of speed and memory efficiency across multiple scenes [19]. MSTransformer [20] pioneers a streamlined one-stage MS-APR approach with transformer architecture. It leverages the transformer decoder not only to improve memory efficiency but also to enhance accuracy significantly.

However, we point out that the learning capacity of transformer encoders is underutilized. As shown in Tab. 1, MSTransformer's encoder self-attention modules do not significantly improve or even degrade performance. Although we discovered low-rank, collapsed attention maps, which are known

---

[†]Corresponding author

38th Conference on Neural Information Processing Systems (NeurIPS 2024).

Table 1: Ablation on encoder self-attention modules in MSTransformer [20]. We report the average of median position and orientation errors for each experiment.

|  | Outdoor [8] | Indoor [27] |
| --- | --- | --- |
| MST [20] | 1.28m/2.73° | 0.18m/7.28° |
| MST w/o encoder SA | 1.21m/2.84° | 0.18m/7.49° |

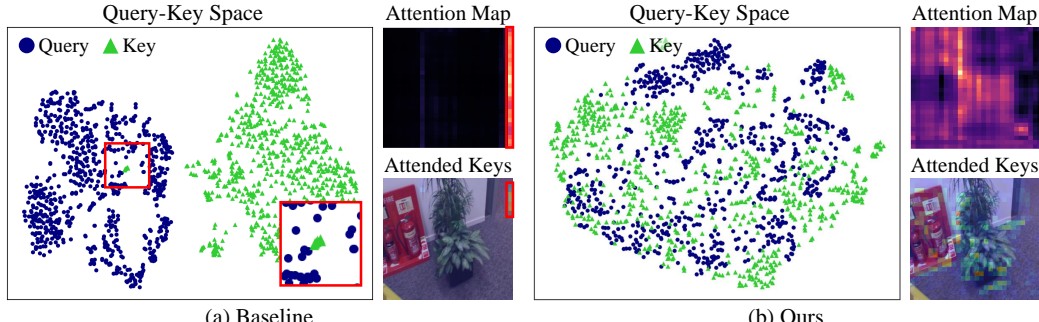

(a) Baseline          (b) Ours

Figure 1: The figure shows query-key spaces, self-attention maps, and attended keys from the orientation transformer encoder of the baseline and ours, respectively. (a) In the case of the baseline, queries and keys are mapped in separate regions, while only a few keys are blended into the query region. Consequently, the self-attention map collapses and whole image features are represented by meaningless few keys, indicating waste of learning capacity of transformer encoder. (b) However, our solution makes queries and keys interact with each other, activating self-attention. This allows the model to obtain crucial global relations within image features, capturing salient global features.

to cause gradient vanishing [21] or training instability [22], general solutions [22–24] were not correct the problem in the case of MSTransformer. This challenge motivates us to elucidate the phenomenon from another perspective in the context of MS-APR.

Therefore, this work proposes brand new analysis: distortion of query-key embedding space, where the attention score originates. We statistically substantiate that queries and keys are mapped in completely separated regions in the embedding space while only a few keys are blended into the query region. In this situation, attention maps inevitably collapse, as all queries are represented similarly to those few keys. Intuitively, it means that all image patches are just represented by only a few image patches, as illustrated in Fig. 1. We conjecture that the task difficulty contributes to the issue; the model should extrapolate the camera pose from a single RGB image across multiple scenes. To support this, we empirically reveal that the model tends to avoid exploring self-relationships at the beginning of the training. These findings align with our additional observations that learnable positional embeddings used in self-attention are also undertrained.

Nevertheless, we expect that image features incorporating global self-relations will be valuable to estimate the camera pose by capturing salient features such as long edges and corners. This work thus introduces simple but effective solutions that activate encoder self-attention modules for MS-APR. Concretely, we design an auxiliary loss which aligns the query region and the key region. By forcing all queries and keys to be mapped into the close and dense space, they are highly encouraged to interact with each other by self-attention—like putting boxers in the ring and blowing the whistle! Furthermore, we explore various positional encoding methods based on empirical evidence since current undertrained one confuses the model to estimate the camera pose with incorrect positions. Finally, fixed 2D sinusoidal encoding is adopted instead of learnable parameter-based methods [25, 26]. This enables the model to consider appropriate positional clues of each image feature during exploring self-relationships from the beginning of the training.

As such, the model can obtain rich global relations from activated self-attention as shown in Fig. 1. These relations are incorporated into image features, acquiring more informative encoder output. Extensive experiments demonstrate that our solution recovers the self-attention successfully by preventing the distortion of query-key space and keeping high capacity of self-attention map [22]. As a result, our model outperforms existing MS-APR methods in both outdoor [8] and indoor [27] scenes without additional memory during inference, upholding the original purpose of MS-APR.

## 2 Related Work

**Absolute Pose Regression (APR)** is to estimate 6-DoF camera pose directly using only an RGB image as input. First proposed by Kendall *et al.* [8], subsequent works [9, 10, 13, 14, 17] have introduced advanced architectures and training methods. However, Sattler *et al.* [28] demonstrated that these models have a tendency to memorize training data, resulting in poor generalization when presented with novel views. In response, IRPNet [16] attempted fast APR by adopting pre-trained image retrieval model as feature extractor. On the other hand, E-PoseNet [15] substituted vanilla CNN with Group-Equivalent CNN to extract geometric-aware image features. Recently, NeRF-based models [11, 12, 29] have improved the APR performance significantly by combining existing APR models with NeRF-W [30] to capture geometric information or synthesize additional images. However, these models have the limitation of being scene-specific, making them impractical for real-world applications [20]. NeRF-based models [11, 12, 29] are particularly vulnerable since NeRF-W is not just scene-specific, but also requires lots of time for taking structure-from-motion, training, and generating additional data.

**Multi-Scene Absolute Pose Regression (MS-APR)** aims to estimate a camera pose from multiple scenes with a single model for scalability. Initially, MSPN [19] made a feature weight database for each scene using CNN, and then fed the indexed features into scene-specific regressors for estimating camera pose in multi-scenes. However, it still had drawbacks of training time and memory inefficiency. Against the backdrop, MSTransformer [20] enabled sound one-stage MS-APR with transformer-based architecture. They proposed to use decoder queries as scene queries to make both scene classification and camera pose regression possible at once. The whole decoder outputs are firstly used to predict the scene index, and then the decoder query corresponding to the scene is used to regress the camera pose. This allows it to become more general through various scenes. Recently, additional studies [31, 32] have been proposed fine-tuning methods, but reverting scene-specific again and using additional parameters. In contrast, we propose simple but effective training methods for training MSTransformer from scratch, maintaining the original purpose of MS-APR.

**Self-Attention** is a attention mechanism which represents input sequences by inter-relationships among all elements. Following the remarkable performance of transformer [33–36], it became widely adopted in various fields. MSTransformer [20] also used self-attention both the transformer encoder and decoder. However, recent studies [21–24, 37–40] have pointed out the failure of self-attention. Firstly, several studies on natural language processing [37, 38] have analyzed that self-attention map collapses to a low-rank matrix, by focusing attention scores to meaningless tokens such as CLS and SEP. In the case of vision, several works [23, 40] have addressed the issue of collapsed self-attention in transformer encoder. There are also studies [21, 22, 24, 39] which fundamentally demonstrated that self-attention maps collapse into low-rank matrices in certain conditions. Similar to these findings, self-attention modules in MSTransformer's encoders do not significantly improve or even impair performance due to the collapse. However, we found that pioneers are not very effective in the case of MS-APR, which is shown in our experiments. Therefore, we analyse the problem from a new viewpoint, distortion of query-key space and undertrained learnable positional embedding, and propose solutions that settle down the problem successfully in MS-APR.

## 3 Preliminary

**Camera Pose.** The location of a captured image $\mathcal{I}$ can be calculated from camera intrinsic and extrinsic parameters. The former is specific to the camera itself, and the latter corresponds to the pose of the camera with respect to a fixed world coordinate system. As such, the camera pose $\mathbf{p}$ can be represented as $(\boldsymbol{t}, \boldsymbol{r})$. Here, $\boldsymbol{t} \in \mathbb{R}^3$ the camera position (translation) and $\boldsymbol{r} \in \mathbb{R}^4$ the camera orientation (rotation). Note that $\boldsymbol{r}$ is an unit quaternion vector so $p$ has 6-DoF.

**Model Architecture.** MSTransformer [20] suggests the model architecture inspired by DETR [35]. It is composed of a CNN, transformer encoder-decoders, a scene classifier, and regressors. Firstly, given an image $\mathcal{I}$, CNN features for the orientation $\boldsymbol{f_r}$ is extracted from middle level while those for the position $\boldsymbol{f_t}$ is computed from middle-high level. $\boldsymbol{f_t}$ and $\boldsymbol{f_r}$ are projected by $1 \times 1$ convolution and reshaped as $\tilde{\boldsymbol{f_t}} \in \mathbb{R}^{N_t \times D}$ and $\tilde{\boldsymbol{f_r}} \in \mathbb{R}^{N_r \times D}$ to be transformer-compatible inputs, where $N_t$ and $N_r$ are the number of tokens for the position transformer and the orientation transformer, respectively.

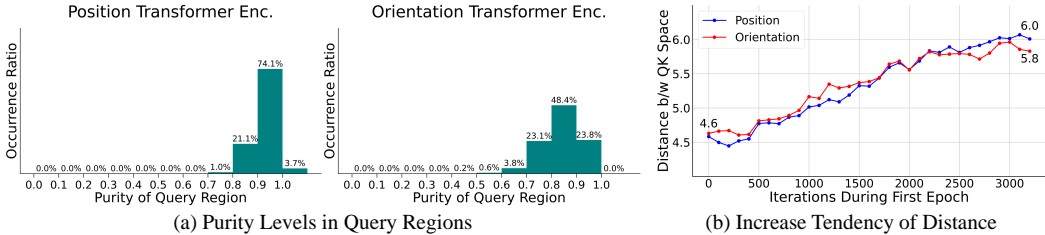

Figure 2: (a) shows the purity levels in query regions with the baseline on 7Scenes, referring the Eq. 2. Note that the purity is 1.0 when the query region is composed only with queries, but slightly lower than 1.0 when a small subset of keys resides in the query region. According to (a), statistical evidence supports the prevalent occurrence of the blending of a few keys into the query region across the entire dataset, both in the position and orientation transformer encoders. (b) illustrates the increasing tendency of distance between the query region and the key region in the encoder. They lean away each other even at the beginning of the training. Here, the distance between query region and key region is an average value across layers and heads.

The architecture of transformer encoder-decoder follows the general transformer [33, 35]. It utilizes decoder query as scene query to make it possible for both scene classification and camera pose regression at once. There are the position and the orientation transformers, each of which has scene queries $z_t \in \mathbb{R}^{M \times D}$ and $z_r \in \mathbb{R}^{M \times D}$ where $M$ is the number of scenes, respectively. In the encoder, there are several layers to reinforce the CNN features. Every layer of the encoder consists of a multi-head self-attention module and multi-layer perceptron (MLP) with layer normalization and residual connection.

Meanwhile, the decoder is designed to not only enhance the decoder queries but also learn the relationship between encoder features and decoder queries. It has multiple layers like the encoder but additionally includes a multi-head cross-attention module in each layer. Afterwards, the scene classifier predicts the scene index $m$ from concatenated decoder outputs $z = [z_t; z_r] \in \mathbb{R}^{M \times 2D}$. In the end, $\hat{t}$ and $\hat{r}$ are predicted by MLP regressors from the $m$-th decoder outputs $z_t{}^m \in \mathbb{R}^D$ and $z_r{}^m \in \mathbb{R}^D$, respectively.

**Self-Attention.** There are three main components in attention mechanism: query $Q$, key $K$, and value $V$. The final output is computed as a weighted sum of $V$, with weights derived from the similarity score between $Q$ and $K$. Formally, given the input $X \in \mathbb{R}^{N \times D}$ with $N$ tokens of dimension $D$, the $h$-th attention head is defined as follows:

$$\text{Attn}_h(X) := A_h V_h,$$

$$\text{where } A_h = \text{softmax}\left(\frac{Q_h K_h{}^\top}{\sqrt{D/H}}\right). \tag{1}$$

Here, $A_h$ is $h$-th attention map and $H$ is the number of heads. When inputs to query, key, and value are all the same, it is usually called as self-attention.

## 4 Problem Analysis

### 4.1 Distortion of Query-Key Embedding Space

We introduce our viewpoint why encoder self-attention modules in MSTransformer are not utilized as shown in Tab. 1. Our main intuition is that the model tends to avoid the self-attention mechanism [38, 24] by distorting the query-key embedding space due to the learning difficulty. On the face of it, collapsed self-attention maps may seem to be the root cause. However, we further investigate the issue, focusing on the query-key embedding space. The attention map is determined by computed similarity scores between queries and keys, as expressed in Eq. 1. In other words, the projection matrices and bias are what the model learns, and the attention map is merely a calculated consequence. As such, the collapsed attention map which induces the model's bypassing comes from the distorted query-key space, therefore we highlight it to address the problem.

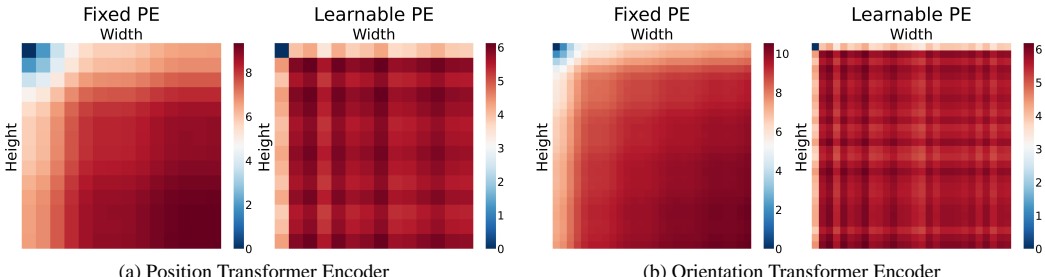

(a) Position Transformer Encoder       (b) Orientation Transformer Encoder

Figure 3: The figure shows L2 distances between the top-left token and other tokens based on the fixed 2D sinusoidal positional encoding and learnable positional embedding, respectively. Here, the learnable positional embedding is the result of training with the baseline. The fixed positional encoding preserves the order of input sequences, but in the case of the learnable positional embedding, tokens not aligned at the same height or width were all treated randomly further away.

When collapsing, queries and keys are completely separated while a small subset of keys are blended into the query region across all layers and heads, both in the position and the orientation encoders of the baseline [20]. To gauge the statistical incidence of such cases, we first clarify the query region and the key region in the query-key embedding space. By clustering $Q \cup K$ through k-means using a mean vector of queries $\bar{q}$ and a mean vector of keys $\bar{k}$ as initial centers, query-dominant region $\hat{Q}$ is obtained from the cluster with center $\bar{q}$. Then we define the purity of query region $\mathcal{P}$ as follows:

$$\mathcal{P} = \frac{1}{|\hat{Q}|} |\hat{Q} \cap Q|. \tag{2}$$

Intuitively, $\mathcal{P}$ is 1.0 when the query region is composed only with queries, slightly lower than 1.0 when small subset of keys are blended into the query region, and about 0.5 when queries and keys are mixed together. Fig. 2 (a) illustrates the statistical results on 7Scenes [27] dataset. It shows that the phenomenon we point out is predominant throughout the whole dataset both for the position and orientation transformer encoders. Under this condition, the attention map inevitably collapses as all queries are computed to be similar to those keys.

We also verify that queries and keys lean away each other from the beginning of the training as shown in Fig. 2 (b). The model is trained to locate only a few keys close to the query region while keeping the long distance between the query region and the key region. Here, the distance is calculated using the L2 metric between their centers. These investigations imply that the model avoids the process of identifying self-relations due to the task difficulty only with the original task loss.

## 4.2 Undertrained Positional Embedding

Moreover, we discuss the importance of appropriate positional encoding for the task and the negative effects of the current positional embeddings in this section. As self-attention is a permutation-invariant mechanism, positional encoding is widely adopted to the queries and keys to account for the order of the input sequence [34–36]. It plays a vital role to utilize the encoder self-attention modules, particularly in MS-APR. To be specific, it is not only important to detect features like roofs and windows in the image, but also crucial to know where they are located in the image for accurate pose estimation. In other words, the model has to estimate the camera pose with shuffled image patches if it uses inappropriate positional embedding.

Despite its importance, learnable positional embeddings in the baseline remain undertrained with deactivated self-attention modules. Fig. 3 shows the visualization results of the distances between the top-left token and other tokens, based on the fixed 2D sinusoidal positional encoding and the learnable positional embedding trained in the baseline. One can observe that distances between tokens are drastically larger if the token has different vertical or horizontal position with the top-left token, in the case of learnable positional embeddings. This makes it even harder for the model to learn geometric relationships between image features by self-attention mechanism.

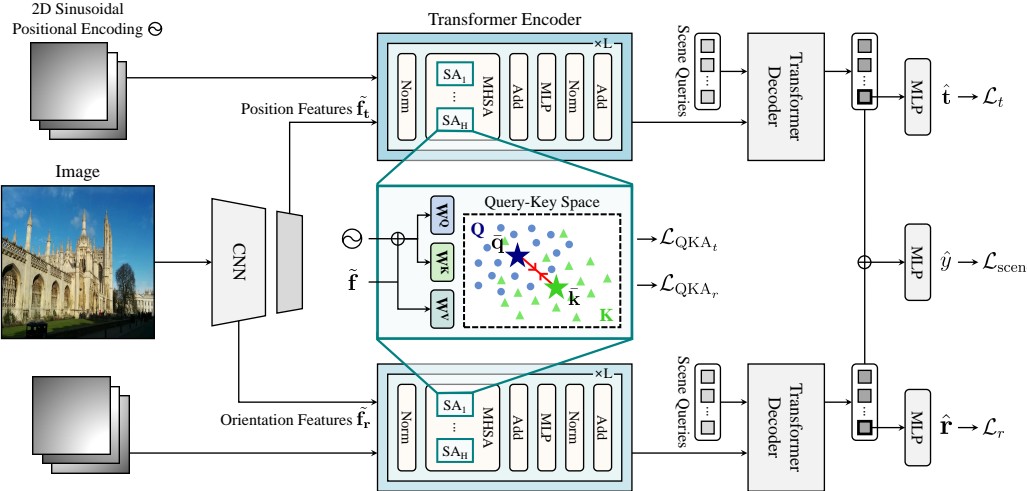

Figure 4: Fig. 4 illustrates the training pipeline with our solutions. We apply additional objectives $\mathcal{L}_{\text{QKA}_t}$ and $\mathcal{L}_{\text{QKA}_r}$ to the model to activate the self-attention modules. Specifically, queries $Q$ and keys $K$ interact with each other by forcing the centroid of query region $\bar{\mathbf{q}}$ and the centroid of key region $\bar{\mathbf{k}}$ to become closer. Here, we encode all input queries and keys with fixed 2D sinusoidal positional encoding to ensure active interaction between $Q$ and $K$ with reliable positional clues.

## 5  Activating Self-Attention for MS-APR

In this section, we introduce our solutions with full objectives for training our transformer-based MS-APR model successfully by mitigating self-attention map collapse. Firstly, we employ the L2 loss to regress the camera position and orientation. Let us denote the ground-truth camera pose as $(\boldsymbol{t}, \boldsymbol{r})$ and the estimated camera pose as $(\hat{\boldsymbol{t}}, \hat{\boldsymbol{r}})$. Then, the position loss and the orientation loss are defined as $\mathcal{L}_t = \|\boldsymbol{t} - \hat{\boldsymbol{t}}\|_2$ and $\mathcal{L}_r = \|\boldsymbol{r} - \frac{\hat{\boldsymbol{r}}}{\|\hat{\boldsymbol{r}}\|}\|_2$, respectively. Here, it is required to balance two losses to make the model learn both camera position and orientation since the former is a scalar value but the latter is a unit quaternion [8, 14]. According to [14], the final pose loss is given by:

$$\mathcal{L}_{\text{pose}} = \mathcal{L}_t \exp\left(-s_t\right) + s_t + \mathcal{L}_r \exp\left(-s_r\right) + s_r, \tag{3}$$

where $s_t$ and $s_r$ are the learnable parameters to adjust the uncertainty.

Secondly, the model should be capable of classifying the scene from the input image to work with multi-scene dataset. Let us denote $y \in \mathbb{R}^M$ as one-hot vector encoding the ground-truth scene index of the input image, and $\hat{y} \in \mathbb{R}^M$ as the predicted probability distribution of scene classes from concatenated decoder outputs $\boldsymbol{z}$. Then, the scene classification loss proposed in [20] is given by:

$$\mathcal{L}_{\text{scene}} = -\sum_{j=1}^{M} y_j \log \hat{y}_j. \tag{4}$$

On top of it, we introduce an auxiliary loss to prevent the model from avoiding the self-attention as depicted in Fig. 4. As we discussed in Sec. 4.1, it is essential to ensure that the queries and the keys do not become alienated from each other while encouraging their interaction by embedding in close proximity. To achieve this, we propose the Query-Key Alignment (QKA) loss as an additional objective. It is defined as follows:

$$\mathcal{L}_{\text{QKA}} = \frac{1}{L} \frac{1}{H} \sum_{l=1}^{L} \sum_{h=1}^{H} \|\bar{\boldsymbol{q}}_h^l - \bar{\boldsymbol{k}}_h^l\|_2, \tag{5}$$

where $L$ is the number of encoder layers, $H$ is the number of heads, and $\bar{\boldsymbol{q}}_h^l$ and $\bar{\boldsymbol{k}}_h^l$ are the mean vectors of the queries and the keys per each encoder layer and head, respectively. By inducing queries and keys to be placed in a shared compact area, we ensure that they interact with each other in the

Table 2: Comparative analysis of MS-APRs on the Cambridge Landmarks dataset (outdoor localization) [8]. We report the median position/orientation errors in meters/degrees.

| Method | King's College | Old Hospital | Shop Facade | St. Mary | Average |
|---|---|---|---|---|---|
| MSPN [19] | 1.73/3.65 | 2.55/4.05 | 2.92/7.49 | 2.67/6.18 | 2.47/5.34 |
| MST [20] | **0.83**/1.47 | 1.81/2.39 | 0.86/3.07 | 1.62/3.99 | 1.28/2.73 |
| +Ours | 0.88/**1.29** | **1.55**/**1.87** | **0.79**/**2.51** | **1.57**/**3.50** | **1.19**/**2.29** |

Table 3: Comparative analysis of MS-APRs on the 7Scenes dataset (indooor localization) [27]. We report the median position/orientation errors in meters/degrees.

| Method | Chess | Fire | Heads | Office | Pumpkin | Kitchen | Stairs | Average |
|---|---|---|---|---|---|---|---|---|
| MSPN [19] | **0.09**/4.76 | 0.29/10.50 | 0.16/13.10 | **0.16**/6.80 | 0.19/5.50 | 0.21/6.61 | 0.31/11.63 | 0.20/8.41 |
| MST [20] | 0.11/4.66 | **0.24**/9.60 | **0.14**/12.19 | 0.17/5.66 | 0.18/4.44 | **0.17**/5.94 | 0.26/8.45 | 0.18/7.28 |
| +Ours | 0.10/**4.15** | **0.24**/**8.79** | **0.14**/**11.59** | 0.17/**5.28** | **0.17**/**3.48** | **0.17**/**5.62** | **0.22**/**7.58** | **0.17**/**6.64** |

self-attention module. Here, we encode all input queries and keys with fixed 2D sinusoidal positional encoding instead of undertrained learnable positional embedding. This guides the model to stably learn self-relationships from the beginning of the training. As a result, the accurate interaction can be accomplished by reflecting the verified correct positional information to input queries and keys.

We apply our QKA loss to both the position and orientation transformer encoders, while adjusting the weight of the loss to integrate with the original task loss. Accordingly, the final auxiliary loss $\mathcal{L}_{\text{aux}} = \lambda_{\text{aux}} \left( \mathcal{L}_{\text{QKA}_t} + \mathcal{L}_{\text{QKA}_r} \right)$, where $\lambda_{\text{aux}}$ is the weight of the auxiliary loss. $\mathcal{L}_{\text{QKA}_t}$ and $\mathcal{L}_{\text{QKA}_r}$ are QKA losses for the position and orientation transformer encoders, respectively. Putting all together, our full objective $\mathcal{L} = \mathcal{L}_{\text{pose}} + \mathcal{L}_{\text{scene}} + \mathcal{L}_{\text{aux}}$.

# 6 Experimental Results

## 6.1 Experimental Setup

**Datasets.** We train and evaluate the model on outdoor and indoor datasets [8, 27], which include RGB images labeled with 6-DoF camera poses. Firstly, we use the Cambridge Landmarks which consist of six outdoor scenes scaled from $875m^2$ to $5600m^2$. Each scene contains 200 to 1500 training data. We conduct the experiment on four scenes in Cambridge Landmarks, which are typically adopted for evaluating absolute pose regression [12, 15, 20]. On the other hand, we use the 7Scenes dataset which consists of seven indoor scenes scaled from $1m^2$ to $18m^2$. Each scene includes from 1000 to 7000 images.

**Training Details.** We entirely follow the configuration of the baseline [20]. We train the model with a single RTX3090 GPU, Adam optimizer with $\beta_1 = 0.9, \beta_2 = 0.999, \epsilon = 10^{-10}$, and the batch size of 8. For the 7Scenes dataset, we train the model for 30 epochs with the initial learning rate of $1 \times 10^{-4}$, reducing the learning rate by $1/10$ every 10 epochs. In the case of Cambridge Landmarks dataset, we train the model for 500 epochs with the initial learning rate of $1 \times 10^{-4}$, reducing the learning rate by $1/10$ every 200 epochs. Afterwards, we freeze the CNN and the orientation branch, then fine-tune the position branch as same as the baseline [20]. The model is trained for 60 epochs with the initial learning rate $1 \times 10^{-4}$, reducing the learning rate by $1/10$ every 20 epochs.

The learnable parameters $s_t$ and $s_r$ for the original task loss are initialized as in [14]. Both for the position and orientation transformer encoder-decoder, the number of layers $L$ is 6 and the number of heads $H$ is 8. Lastly, we set the $\lambda_{\text{aux}}$ for our query-key alignment loss as 0.1.

## 6.2 Comparative Analysis

**MS-APR Methods.** We compare the model trained our solutions with other single-frame multi-scene APR models for fair comparison, *i.e.*, excluding scene-specific methods. Tab. 2, Tab. 3, and Tab. 4 show the experimental results on four outdoor scenes in Cambridge Landmarks [8] and seven indoor scenes in 7Scenes [27], respectively. With our solutions, the model shows better performance through all scenes, compared to other MS-APR methods. This results verify that activating self-attention has

Table 4: Comparative analysis of the baseline and ours. We report the localization recall at several thresholds on Cambridge Landmarks and 7Scenes datasets.

| Method | Cambridge Landmarks [8] | | | | 7Scenes [27] | | | |
|---|---|---|---|---|---|---|---|---|
| | $(1m, 5°)$ | $(1m, 10°)$ | $(2m, 5°)$ | $(2m, 10°)$ | $(0.2m, 5°)$ | $(0.2m, 10°)$ | $(0.3m, 5°)$ | $(0.3m, 10°)$ |
| MST [20] | 32.6 | 35.8 | 60.4 | 68.2 | 28.8 | 50.2 | 34.4 | 63.5 |
| +Ours | **35.8** | **38.5** | **65.5** | **72.7** | **32.6** | **52.6** | **39.5** | **67.1** |

Table 5: Comparison with alternative methods for collapsed SA and positional encoding methods. We report the average of the median position/orientation errors on 7scenes dataset.

(a) Alternative Methods for Collapsed SA

| Method | 7Scenes [27] |
|---|---|
| Improved SN [23] | $0.18m/7.04°$ |
| $1/\sqrt{L}$-scaling [24] | $0.18m/6.87°$ |
| $\sigma$Reparam [22] | $0.19m/6.81°$ |
| QK Alignment | $\mathbf{0.17}m/\mathbf{6.64°}$ |

(b) Alternative Positional Encoding Methods

| Method | 7Scenes [27] |
|---|---|
| T5 PE [25] | $0.18m/6.97°$ |
| Rotary PE [26] | $0.18m/6.94°$ |
| Fixed PE | $\mathbf{0.17}m/\mathbf{6.64°}$ |

the potential to become a general module, generating task-relevant features across multiple scenes, regardless of the specific features of each scene.

**Alternative Methods for Collapsed Self-Attention.** We apply general-purpose techniques [22–24], proved to alleviate collapsed self-attention, to the baseline to compare the effectiveness of each method and ours in MS-APR. Here, we conduct experiments with fixed positional encoding, as incorrect inputs may suppress the potential of the solutions. Tab. 5a shows that existing techniques have little or no effect in terms of recovering the self-attention for the task. Meanwhile, ours significantly improves the baseline in both conditions. We conjecture that it is required to guide the model in a more direct manner to utilize self-attention for MS-APR.

**Alternative Positional Encoding Methods.** We conduct experiments with recent advanced learnable positional encoding methods [25, 26] with our QKA loss. As shown in Tab. 5b, even advanced methods are not suitable to the task as they mainly depend on learnable parameters and relative position. We assume that absolute, purified positional clues are important to stabilize the training of the embedding space in the case of MS-APR.

## 6.3 Quantitative Analysis

**Attention Entropy of Self-Attention Maps.** To provide quantitative evidence that our solutions enhance performance of model by activating the encoder self-attention, we measure attention entropy [22] defined as follows:

$$\text{Ent}(\text{Attn}(\boldsymbol{X})) = \frac{1}{H} \sum_{h=1}^{H} \text{Ent}(\boldsymbol{A}_h), \tag{6}$$

$$\text{where } \text{Ent}(\boldsymbol{A}_h) = \frac{1}{N} \sum_{i=1}^{N} \left( -\sum_{j=1}^{N} a_{ij} \log (a_{ij}) \right). \tag{7}$$

Note that high attention entropy indicates training stability and high representation capacity of self-attention map [22].

Fig. 5 shows the attention entropy for each encoder layer in both the position and orientation transformers. As illustrated, the model employing our solutions has significantly higher attention entropy than the original baseline model across all encoder layers. These results validate that our solutions successfully induce the transformer-based model to activate the encoder self-attention, thus improving the representation quality of self-attention map.

**Purity Levels in Query Regions.** To verify the effectiveness of our auxiliary loss on rectifying the distortion of query-key space, we measure the purity levels in query regions defined in Eq. 2. The statistical results for both the baseline and the model employing our solutions are presented in Fig. 6.

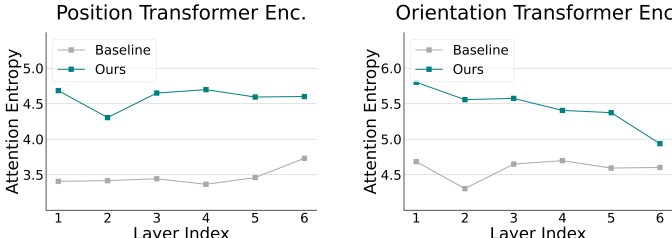

Figure 5: The figure shows the attention entropy of the baseline and ours for each encoder layer. It demonstrates that our solutions significantly improve the utilization of encoder's learning capacity.

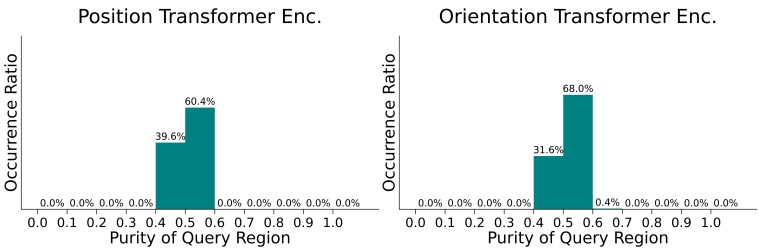

Figure 6: The results show the change of purity of query region $\mathcal{P}$, defined in Eq. 2, across 7Scenes dataset. Note that the purity is 1.0 when the query region is composed only with queries, but slightly lower than 1.0 when few keys resides in the query region. Compared to the baseline, there are no cases where only a few keys are blended into the query region with our solution.

Here, the purity levels represent the proportion of the queries allocated to the query-dominant cluster. Our observations reveal that the nearly half keys belong to the query region in almost all cases, indicating that the query and key regions are highly interleaved. This demonstrates that our solutions restrain all queries from being treated as similar to only a few keys, thus preventing self-attention collapse and activating self-attention.

### 6.4 Qualitative Analysis

Fig. 7 reports visualization of query-key space by t-SNE and attention results of the baseline and the model employing our solution. Note that the attention results depict the scores of the keys, averaged over the queries in the self-attention map. One can observe that the query regions and key regions of the baseline are separate. In contrast, the regions are highly aligned by our solutions, reducing the occurrence of only a few keys blended into the query region. This condition necessitates that the model reflects global interactions between queries and keys, therefore, global relations are incorporated into image features as depicted in Fig. 7. We can see that geometric relations such as long edges are usually obtained, which are the critical points for the task. More examples are in the supplementary material.

### 6.5 Ablation Study

We conduct an ablation study on our solutions to verify the effectiveness of each component. As shown in Tab. 6, our QKA loss enhances the performance in both the outdoor and indoor datasets. Additionally, overall performances are further improved with fixed positional encoding, rectifying the incorrect inputs in the self-attention. This ablation study indicates that several useful self-relation can be extracted from noisy content features by self-attention, but reliable positional clues are particularly important to learn various crucial relationship for estimating more accurate camera pose.

## 7 Discussion

**Limitation and Future Work.** The proposed auxiliary loss is designed based on the assumption that an image has many key features which are useful for estimating the camera pose. However, there

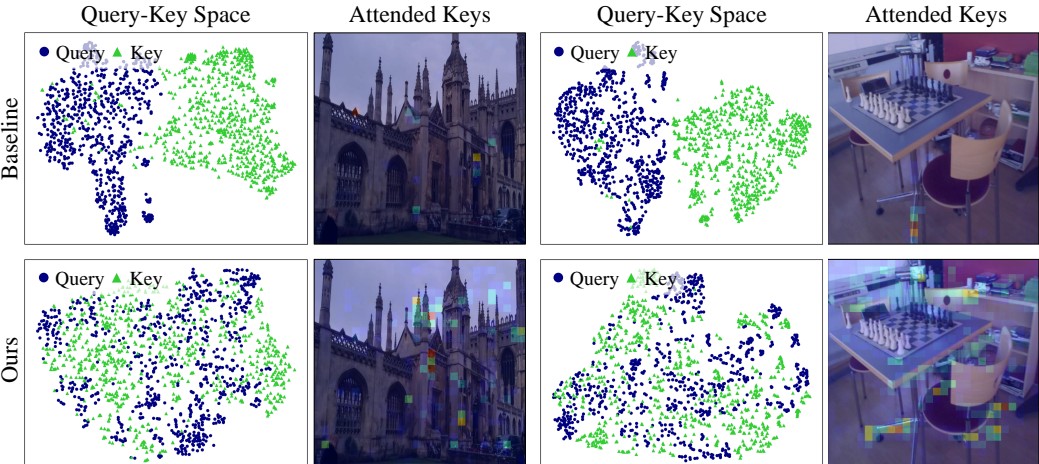

Figure 7: The figure shows the examples of t-SNE results of query-key space and attention results of the baseline and the model employing our solution. The query and key regions are separate in the baseline while our solutions align the regions, enabling the model to focus on salient global features and incorporate self-relations into image features.

Table 6: Ablation on our solutions. We report the average of the median position/orientation errors.

| Method | Outdoor [8] | Indoor [27] |
|---|---|---|
| MST [20] | 1.28m/2.73° | 0.18m/7.28° |
| MST+Fixed PE | 1.29m/2.46° | 0.18m/6.80° |
| MST+QKA | 1.22m/2.68° | 0.18m/7.09° |
| MST+Fixed PE+QKA | **1.19**m/**2.29**° | **0.17**m/**6.64**° |

could be a side effect if only a few key features are in the image; *i.e.*, if a dynamic moving object mainly occupies the image. Developing an algorithm to decide whether global self-attention is useful for the image or not could be an interesting future work.

**Broader Impacts.** The camera pose estimation technology with this work can be expected to have several positive social impacts, such as being utilized in programs to locate missing persons. On the other hand, it could potentially be misused for purposes such as illegal location tracking programs.

## 8 Conclusion

This work has exposed that self-attention modules in transformer encoders are not activated in MS-APR. Looking beyond the collapsed self-attention maps, we have discovered the distortion of query-key embedding space; queries and keys are entirely separate while a small subset of keys resides in the query region. We have examined the predominance of the phenomenon, adducing statistical and empirical evidence. Additionally, the importance of accounting for the order of the input sequence has been addressed in the context of the task. Based on these analyses, this work has proposed simple but effective solutions to activate self-attention by rectifying the distorted query-key space with alignment loss and appropriate positional clues. As a result, the model could learn salient global features through self-attention, thus achieving state-of-the-art performance both in outdoor and indoor scenes without additional memory in inference. Our extensive experiments have demonstrated the effectiveness of proposed solutions in terms of recovering self-attention for MS-APR.

## Acknowledgement

This work was supported in part by MSIT&KNPA/KIPoT (Police Lab 2.0, No. 210121M06), MSIT/IITP (No. 2022-0-00680, 2020-0-01821, 2019-0-00421, RS-2024-00459618, RS-2024-00360227, RS-2024-00437102, RS-2024-00437633), and MSIT/NRF (No. RS-2024-00357729).

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

# A Appendix

## A.1 Training Details

We entirely follow the setting of the baseline [20]. The settings are as below.

**Data Augmentation.** Data augmentation is performed in the same way as [8], which is also adopted in the baseline. Afterwards, brightness, contrast, and saturation are randomly jittered for photometric robustness. Test images are also resized to $256 \times 256$, but they are center-cropped to $224 \times 224$, without any photometric augmentation. Additionally, following the baseline, smaller scenes in the Cambridge Landmarks dataset [8] are oversampled during training to deal with data imbalance problem.

**Backbone.** As did in the baseline [20], we utilize a CNN model, the EfficientNet-B0 [41], to extract local image features from input images. The image features $f_t$ for the position and $f_r$ for the orientation are extracted from the fourth block and the third block, respectively. Accordingly, the size of $f_t$ is $14 \times 14 \times 112$ and the size of $f_r$ is $28 \times 28 \times 40$, *i.e.*, the number of transformer encoder tokens $N_t$ and $N_r$ are 196 and 784, respectively.

## A.2 Visualization

**Query-Key Spaces.** In addition to the statistical and quantitative results introduced in the main paper, we visualize the query-key spaces across heads and layers. Fig. A1 reports the visualization results of query-key spaces from the baseline [20] across all even layers and heads. It qualitatively shows that the distortion of query-key space occurs in most heads in transformer encoder of the baseline. In contrast, one can observe that queries and keys properly intertwine each other with our solutions as shown in Fig. A2.

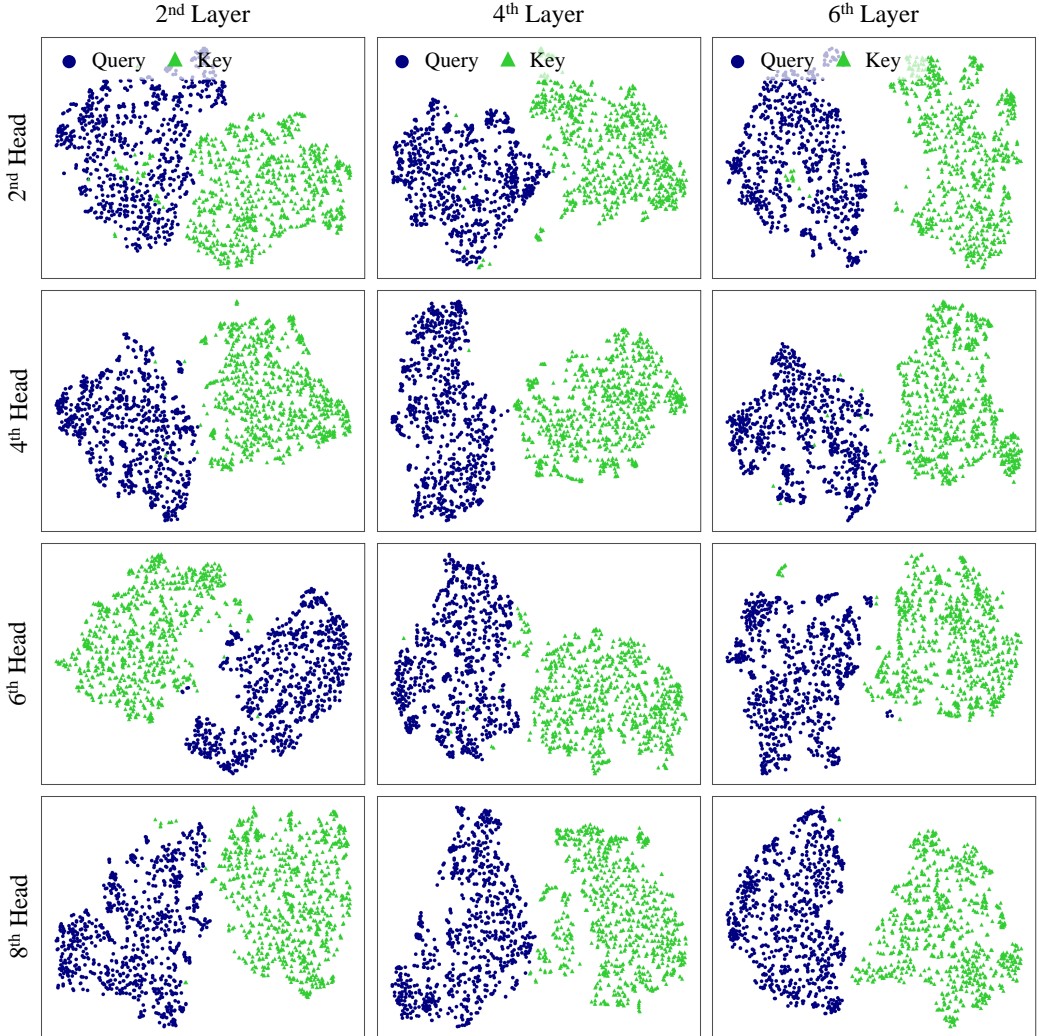

Figure A1: The figure shows the t-SNE results of query-key space from the baseline across even layers and heads. Overall, query and key regions are separate while a small subset of keys are blended into the query region.

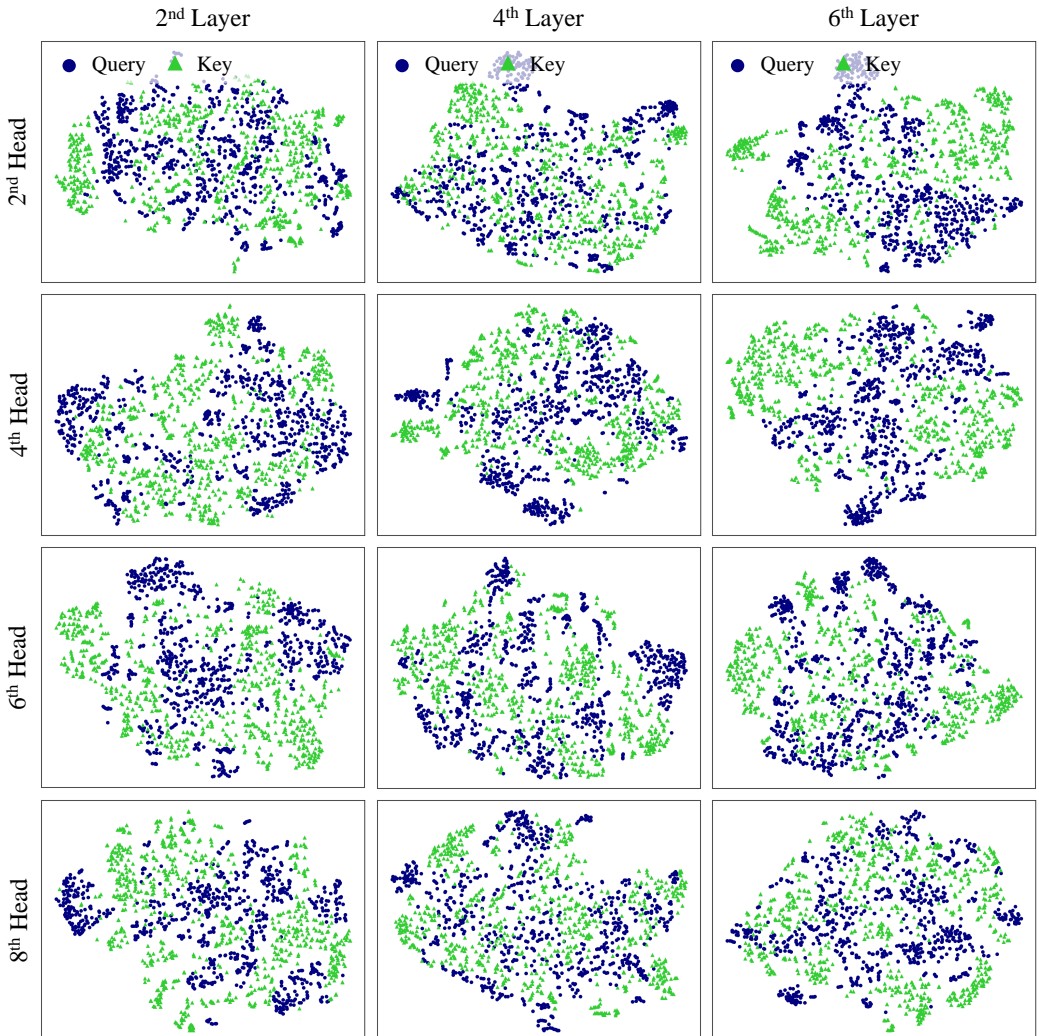

Figure A2: The figure shows the t-SNE results of query-key space from the model with our solutions across even layers and heads. The problem we point out is resolved with our solutions; similar query subsets and key subsets are grouped together.

