# OpenReview forum: "Activating Self-Attention for Multi-Scene Absolute Pose Regression"
_NeurIPS.cc/2024/Conference — NeurIPS 2024 poster_

### Official Review · Reviewer_AzjT · 2024-07-06

**Soundness:** 2
**Presentation:** 3
**Contribution:** 2
**Rating:** 5
**Confidence:** 4

**Summary:**

In this paper, the authors focus on improving the performance of multi-scene absolute pose regression models based on transformers. From statical analysis, the authors assume that the distortion between Q and K features in self-attention and the learnable position encoders are the reasons. Therefore, a Q-K alignment loss and the fixed-pose encoding method are adopted and experiments demonstrate the efficacy on indoor 7scenes and CambridgeLandmarks datasets.

**Strengths:**

The strengths of this paper are as follows.

1.	Originality. The undistortion of Q-K features in self-attention is not a new direction as also mentioned in the related works section. While this paper may be the first one to apply it to the multi-scene absolute pose regression task and approve the efficacy. Besides, the authors also demonstrate that the fixed-position encoding works better than the learnable position encoding because learnable position encoding breaks the order the input sequences. These contributions are useful for improving the accuracy of the pose regression as shown in the experiments.

2.	Quality. The proposed algorithm is easy to follow, and the paper is well written.

**Weaknesses:**

The proposed method is easy to follow as I mentioned before, so I don’t have many concerns. Several minor concerns are as follows.

1.	Contribution. I can understand this paper starts from analyzing the application of the absolute pose regression, but the key technique about the distortion of Q-K features in self-attention mechanism comes from previous works [19]. As this is the major contribution of this paper, I am not very sure if it is enough. Although an additional Q-K alignment strategy is proposed as an additional contribution, its improvement over prior strategies such as [19] is not significant (0.02m and 0.17 deg). These improves may come from a suitable learning rate or more proper hyper-parameters balancing different losses.

2.	Results. Overall, the improvements against the baseline method MSTransformer [17] on indoor (0.02m, 0.64deg) and outdoor (0.09m, 0.44deg) datasets are not significant, which further degrades the contribution of this paper.

**Questions:**

Please see the weakness.

**Limitations:**

Yes.

---

> ### Author Rebuttal · Authors · 2024-08-06
>
> **Q1. Distinctive contribution**
>
> A1. We can understand why this might be a concern.
> [19] and our research both address the underlying problem of attention collapse.
> However, [19] and ours are **clearly different in terms of the analysis on the causes and the solution for the attention collapse.**
> Firstly, [19] points out that the large gradient norm and increasing spectral norm in the early layers of transformers induce the collapse with training instability and degraded performance.
> Hence, they propose a method (σReparam) for regularizing the gradient and spectral norm with pre-layer normalization.
>
> On the other hand, we identify the distortion of the query-key space and the undertrained positional encoding as the main issues.
> That is, when the queries and keys are mapped into completely different spaces while only a few keys exist in the query space, the attention collapses and the components in self-attention modules are deactivated.
> Not just measuring attention entropy like [19], we also **define the new index, purity, to statistically demonstrate the predominance of this issue across the APR dataset.**
> Finally, we introduce an auxiliary loss which aligns the query and key regions as well as fixed positional encoding.
> Therefore, the major contribution stands apart from [19], and consideration of the novel contributions it offers would be appreciated.
>
> **Q2. Performance improvement**
>
> A2. We understand that the improvements may not appear significant at first glance.
> However, please take into account two hidden aspects regarding the conventional evaluation of median errors in position and orientation in APR.
> Firstly, due to the small numerical values, the errors tend to appear minor. Despite this, **our method reduces the position/orientation error rates by 7%/17% in outdoor settings and by 6%/9% in indoor settings.**
>
> Secondly, median error, being a *median*, does not fully reflect the overall performance of the model.
> To address this, we compared the performance with the baseline using the recall evaluation metric, which is widely adopted in 2D-3D correspondence-based camera pose estimation.
> As shown in Table A1 on the global rebuttal page, **our method achieves 3-5%p higher recall across various thresholds and datasets compared to the baseline.**
> Particularly, there are substantial improvements of outliers, which are not evident in the median error metrics.
> By forcing the model to find helpful cues for the task, it becomes especially effective in leveraging previously challenging edge cases.
> In conclusion, we would emphasize our method's superiority and the remarkable performance improvements it provides over the baseline.
>
> **Q3. Performance difference between [19] and ours**
>
> A3. First of all, please allow us to clarify that there is a trade-off between position and orientation [6, 12].
> Thus, it is hard to see that [19] improves performance on the 7Scenes dataset since the median orientation error is decreased but the position error does not.
> In addition, we report the comparative analysis between [19] and our QKA loss on Cambridge Landmark dataset in Table A5 of the global rebuttal page.
> As the scale of the scene increases, it can be observed that our method shows a clear difference in median error compared to [19].
> As mentioned earlier, median error may not be as visually striking as other evaluation metrics; however, please take into account that such a difference in median error is not marginal.

---

> > ### Author Response · Authors · 2024-08-14
> > **Response Uploaded**
> >
> > Thank you for taking the time and effort to review our rebuttal.
> >
> > We have uploaded our responses according to the reviewer's request.
> >
> > With only one hour left for discussion, we kindly ask you to verify our responses.
> >
> > Your verification will help us improve the overall quality and clarity of our work.
> >
> > If we have satisfactorily addressed your concerns, we would appreciate a positive reassessment.

---

> > > ### Comment · Reviewer_AzjT · 2024-08-14
> > > **post rebuttal**
> > >
> > > Thanks for the updating. I don't have any other concerns. I stick to my initial rating.

---

### Official Review · Reviewer_FKQN · 2024-07-08

**Soundness:** 2
**Presentation:** 3
**Contribution:** 2
**Rating:** 4
**Confidence:** 4

**Summary:**

The paper investigates multi-scene absolute pose regression from a new perspective: query-key embedding space. Focusing on distortion of queries and keys, the paper solutions to activate self-attention, which includes an auxiliary loss to align queries and keys and fixed sinusoidal positional encoding. Experimental results demonstrate that the proposed method outperforms existing MS-APR methods in outdoor Cambridge Landmarks and indoor 7 Scenes.

**Strengths:**

1.The paper proposes new analysis that focuses on distortion of query-key embedding space.

2.The paper presents solutions to reduce the distortion that covers an auxiliary loss that aligns queries and keys and fixed sinusoidal positional encoding.

3.Experimental results demonstrate that the proposed method can reduce the localization error.

**Weaknesses:**

1.The motivation of Multi-Scene Absolute Pose Regression is somewhat insufficient. The paper declares that Multi-Scene Absolute Pose Regression can satisfy the needs of speed and memory efficiency across multiple scenes. However, these are not validated in the experiments, especially compared with single-scene methods.

2.The proposed method seems to only support transformer based APR, which are limited to its further applications.

3.Some references are missing. Although the paper focuses on Multi-Scene Absolute Pose Regression, the references about single-scene camera relocalization methods still should be discussed, including single-scene ARP methods and 2D-3D correspondence based methods.

4.The experimental results are limited, reflected by the following aspects.

(1)The paper only lists the localization results in comparison with transformer based Multi-Scene APR methods, but the comparisons with other single-scene state-of-the-art methods are missing, especially the speed and memory efficiency which is declared as advantages of Multi-Scene methods.

(2)From Tables 1,2, the improvements of the proposed method seem not obvious, which can not show the method superiority. More discussions are preferred.

**Questions:**

1.Currently, the 2D-3D correspondence based methods (also called coordinate regression based methods) still achieve the state-of-the-art localization performance in both static and dynamic scenes, such DSAC*, KFNet and so on. It is curious that whether the proposed method can apply to 2D-3D correspondence based methods?

2.Does the proposed method affect the network training convergence time?

**Limitations:**

The authors have adequately addressed the limitation.

---

> ### Author Rebuttal · Authors · 2024-08-06
>
> **Q1. Speed & memory efficiency**
>
> A1. We would like to bring to your attention that the advantages of MS-APR mentioned have already been claimed and proven by previous research, and are not new assertions from our side.
> Firstly, it is known that APR methods are much faster and more memory-efficient than 2D-3D correspondence-based methods because they do not require 3D point clouds or RANSAC algorithms [6, 13, 16, 17].
> APR methods show almost the same speed; but scene-specific models have limitation of memory inefficiency for large-scale application.
> Hence, MS-APR methods propose new architectures that ensure memory efficiency for multiple scenes.
> Besides, our methods do not require additional modules, thereby **maintaining the memory efficiency as the baseline.**
> It is why we focus on performance differences in the experiment section.
> To address the issue, we display the memory requirements of recent single-scene and multi-scene APR methods as the number of scenes increases in Table A3 on the global rebuttal page.
>
> **Q2. Comparison with single-scene methods**
>
> A2. Firstly, allow us to clarify by presenting the localization results in comparison with all MS-APR methods, including MSPN [16], which is a CNN and MLP-based model.
> Secondly, please consider that **to maintain a fair comparison**, we did not include 2D-3D correspondence-based methods or single-scene APR methods.
> For 2D-3D correspondence-based methods, 3D point clouds are required, and thus many APR works [7-10, 13-14, 16-17] have excluded them from the comparison.
> In this context, several recent single-scene APR methods, which adopt NeRF-W that also requires 3D point clouds, are also excluded from the fair comparison.
> Although other single-scene APR methods still pose fairness issues in terms of the number of model parameters, we show the comparison results in Table A4 on the global rebuttal page.
>
> **Q3. Performance improvement**
>
> A3. We understand that the improvements may not appear significant at first glance.
> However, please take into account two hidden aspects regarding the conventional evaluation of median errors in position and orientation in APR.
> Firstly, due to the small numerical values, the errors tend to appear minor. Despite this, **our method reduces the position/orientation error rates by 7%/17% in outdoor settings and by 6%/9% in indoor settings.**
>
> Secondly, median error, being a *median*, does not fully reflect the overall performance of the model.
> To address this, we compared the performance with the baseline using the recall evaluation metric, which is widely adopted in 2D-3D correspondence-based camera pose estimation.
> As shown in Table A1 on the global rebuttal page, **our method achieves 3-5%p higher recall across various thresholds and datasets compared to the baseline.**
> Particularly, there are substantial improvements of outliers, which are not evident in the median error metrics.
> By forcing the model to find helpful cues for the task, it becomes especially effective in leveraging previously challenging edge cases.
> In conclusion, we would emphasize our method's superiority and the remarkable performance improvements it provides over the baseline.
>
> **Q4. Applicability**
>
> A4. We would like to highlight the applicability of our method in two key aspects.
> Firstly, experiments in APR have only been conducted on small datasets, but it will not be long before APR methods need to be evaluated on large-scale datasets such as Aachen Day-Night or RobotCar, which are already tested in 3D-based methods.
> Accordingly, it is worth considering whether scene-specific models will remain competitive in terms of memory requirements as shown in Table A3.
> [16] proved that there is no need for a database in multi-scene by utilizing transformer-based model, drawing the potential of APR.
> Therefore, we believe that **transformer-based APR models will become more prevalent, and our method is applicable to any transformer with deactivated self-attention.**
>
> Furthermore, we would suggest that our analysis and method are **not limited to APR applications.**
> While our research began with addressing attention collapse in APR, our method can also be adopted as a solution for other vision tasks suffering from similar issues.
> To validate this, we applied our method to the temporal action detection task, which uses transformer-based models and exhibits attention collapse.
> As shown in Table A2 and Figure A1 on the global rebuttal page, the baseline DETR struggles with learning self-attention mechanism, and our method significantly improves performance by resolving this issue.
> Therefore, we ask for reconsideration of the strong applicability and scalability of our work in both APR and transformer research.
>
> **Q5. Application to 2D-3D correspondence-based**
>
> A5. That is an interesting question.
> Our method is designed to maximize the activation of image features that assist in camera pose estimation.
> In this context, it could also be applied to 2D-3D correspondence-based methods that aim to reinforce image features through the self-attention mechanism.
> For instance, it could be applied to the ViT Encoder used in [B].
> However, please note that they differ from APR methods in terms of task loss and learning complexity; the attention outcomes might differ from those observed in APR.
>
> [B] Revaud, Jerome, et al. Sacreg: Scene-agnostic coordinate regression for visual localization. CVPR, 2024.
>
> **Q6. Training convergence time**
>
> A6. Thank you for highlighting the new benefit of our research.
> While the baseline model is trained on Cambridge Landmarks for 600 epochs, our model achieved better performance with only 500 epochs of training, thus indicating a clear benefit in terms of training convergence.

---

> > ### Author Response · Authors · 2024-08-14
> > **Response Uploaded**
> >
> > Thank you for taking the time and effort to review our rebuttal.
> >
> > We have uploaded our responses according to the reviewer's request.
> >
> > With only one hour left for discussion, we kindly ask you to verify our responses.
> >
> > Your verification will help us improve the overall quality and clarity of our work.
> >
> > If we have satisfactorily addressed your concerns, we would appreciate a positive reassessment.

---

### Official Review · Reviewer_K2gT · 2024-07-12

**Soundness:** 2
**Presentation:** 3
**Contribution:** 2
**Rating:** 5
**Confidence:** 5

**Summary:**

The paper analyzes the collapse of self-attention map in Multi Scene Pose Transformer model and proposes two simple but effective methods: auxiliary loss and fixed 2D sinusoidal encoding to solve this problem. The improved method delivers SOTA  performance on the Multi Scene Pose Regression task.

**Strengths:**

1. The proposed auxiliary loss seems a good solution for query-key distortion in MS Transformer which is simple and effective.
2. The figures and tables in this paper are exceptionally clear and well-organized, making the paper easy to understand and interpret.

**Weaknesses:**

1. Since query-key distortion is studied in the transformer literature and the fixed 2D sinusoidal positional encoding is an off-the-shelf module, the methods in this paper seems to be lack of novelty. It would therefore be nice to discuss more on why APR task will lead to query-key distortion rather than saying "The model tends to avoid the self-attention mechanism due to the learning difficulty." in line 127-128. The paper reads as if it's discussing improvements to the transformer and not related to APR.

2. Ablation study in Table6 indicates that most of the performance improvements come from the using of fixed sinusoidal positional encoding, a more complete ablation study may be helpful to show the ability of the proposed loss function.

3. Section5: Activating Self-Attention for MS-APR is overly detailed, most of the information is a repetition of MS-Transformer.

**Questions:**

The results in Table5 indicates that the loss function is a very hard constraint that limits the purity of query region to [0.4, 0.6), such constraint is not suitable for all situations. Do authors have comments?

**Limitations:**

The manuscript is limited in scholarship, missing references to more APR methods like [A].

[A] Sc-wls: Towards interpretable feed-forward camera re-localization, ECCV 2022

---

> ### Author Rebuttal · Authors · 2024-08-06
>
> **Q1. Novelty**
>
> A1. We would like to highlight the novelty of our unique analysis and simple but effective solution.
> By focusing on a different aspect from previous studies [18-21, 31-34], we identified the problem in self-attention modules of APR models and solved it, which previous works could not.
> Specifically, they usually explored attention map theoretically to activate self-attention.
> However, as shown in our experimental results, these conventional solutions did not work for APR.
> We think that we provide meaningful discussion by discovering the query-key distortion and **statistically and empirically validating the problem using the dataset specific to this task.**
>
> Positional encoding is no exception.
> Previous studies primarily investigated various learnable positional embeddings to assist self-attention mechanism.
> However, our work found that these variations were not effective for this task and actually increased the difficulty of reasoning with self-attention.
> Although 2D sinusoidal positional encoding is off-the-shelf, it is sufficient to provide the essential position information for self-attention in APR.
>
> We believe that establishing a practical and effective method that improves performance without compromising speed and memory efficiency is one of the most important discussions in MS-APR.
> From this perspective, our methods contribute to MS-APR by activating deactivated modules perfectly **without causing any slowdown or additional memory usage.**
>
> **Q2. Learning difficulty**
>
> A2. Rather than query-key distortion occurring because of estimating camera pose, we hypothesize that the problem stems from the **APR task being difficult due to limited data and the need to extrapolate 3D values from 2D inputs.**
> In other words, self-attention, which has few biases and constraints, may fall back on shortcuts under difficult conditions for reasoning.
> Hence, we suspect that similar issues may arise in other challenging fields with transformer-based models, which struggle with limited data and the need for sophisticated reasoning.
>
> However, it is too difficult to quantify the learning difficulty.
> Therefore, we have explored other demanding vision tasks to verify the effectiveness of our method.
> Please refer to Table A2 and Figure A1 on the global rebuttal page for our experimental results on the DETR-based model in temporal action detection.
> Similar issue of attention collapse in self-attention can be found, and our proposed auxiliary loss effectively solves the problem, leading to significant performance improvements.
>
> **Q3. Ablation study**
>
> A3. Please refer to Table A6 for the complete ablation study on the global rebuttal page.
>
> **Q4. Hard constraint**
>
> A4. Thank you for the great question.
> As we discussed in the limitation, if only specific parts of an image contain information useful for camera pose estimation, such as when a dynamic moving object occupies a large portion of the image, our method might introduce side effects.
> However, these cases are not common.
> Camera pose estimation generally relies on the overall layout of the scene, like the long edges between the ceiling and walls, rather than specific parts of the image.
> Therefore, it is postulated that **applying a hard constraint to activate most features could be more beneficial overall for the task.**
> Namely, we presume that our method is effective for APR as it provides stronger regularization compared to other methods aimed at resolving attention collapse.
>
> **Q5. Missing references**
>
> A5. Thank you for bringing this to our attention.
> We will make sure to include references to [A] and other relevant works in our revised version.

---

> > ### Comment · Reviewer_K2gT · 2024-08-14
> > **BA**
> >
> > I have updated the recommendation to BA.
> >
> > This insight has value in APR, echoing insights from other methods like [A] that attending to useful regions help APR. However, a proper literature discussion is needed to help the APR community converge to useful scientific conclusions.
> >
> > [A] Sc-wls: Towards interpretable feed-forward camera re-localization, ECCV 2022

---

> > > ### Author Response · Authors · 2024-08-14
> > >
> > > We appreciate your positive feedback on the value of our insights in APR and how it echoes other methods like [A].
> > >
> > > We agree that a proper literature discussion is crucial for helping the APR community converge on useful scientific conclusions.
> > >
> > > To address this, we will include a comprehensive discussion, along with references to [A] and other relevant works, in our revised version.

---

### Official Review · Reviewer_aJYL · 2024-07-13

**Soundness:** 4
**Presentation:** 4
**Contribution:** 3
**Rating:** 7
**Confidence:** 5

**Summary:**

This paper is about a improving self attention in the transformer architecture for multi-scene APR. The show that the self attention module in the SOTA transformer model for APR is actually not helping much and offer an potential explanation. The paper claims that the keys and queries end up in different spaces, such that the inner product between keys and queries is very close to zero in most cases, leading the attention to collapse to zero. The paper proposes an additional loss term that encourages the mixing of queries and keys leading to much more overlap. The addition of this term results in a noticeable improvement in APR metrics across indoor and outdoor datasets.

**Strengths:**

+ The motivation for the paper is clear and concise and overall well laid out. Table 1 lists metrics with and without self attention showing the limited utility. It is shown empirically that the keys and queries are isolated form one another as well as theoretically explained why this is an issue. The proposed approach is well explained, straightforward, and has the desired effect of quantitatively improving attention and qualitatively improving APR metrics. This will clearly be used with mulitscene transformer based APR methods going forward because it's simple without requiring any extra data and is effective.

+ The problem analysis Section 4 is thorough and useful. The side effects of the commonly used techniques are clearly explained and the arguments for why this is a problems is compelling.

+ There are sufficient implementation details. I could easily reimplement this paper from the information provided. The hyper parameters are shared with [17] so it is unlikely that the performance gain is due to hyperparameter tweaking.

+ Thorough ablation of different methods for solving the SA problem as described as well as different positional encoding methods. The methods proposed in the paper are validated as being the best on these datasets.

+ This is one of the few papers I've seen using attention maps in pose regression where I feel like the visualizations and discussion around attention are actually meaningful.

**Weaknesses:**

- Table 5 is difficult to parse. I feel like it could easily be represented with histograms as in Figure 2a. Similarly I feel like the histograms a fairly course. The point comes across okay but I'm not sure why such a course histogram would be used.

**Questions:**

In Table 3, MST and +Ours is incorrectly bold for the position error for the Office scene.

**Limitations:**

Yes.

---

> ### Author Rebuttal · Authors · 2024-08-06
>
> **Q1. Histogram**
>
> A1. We thought that using coarse bins was suitable for visualizing and showing the general trend differences in purity between the baseline and our method.
> However, we agree with the reviewer's suggestion to provide a more specific analysis.
> Accordingly, we present the baseline's purity with a fine-level histogram and include the visualization for our method in Figure A2 on the global rebuttal page.
>
> **Q2. Incorrect bold**
>
> A2. Thank you for bringing this to our attention.
> We will make sure to correct it in our revised version.

---

### Author Rebuttal · Authors · 2024-08-06

Thank you for reviewing.

Responses to the questions can be found under each individual review.

The global rebuttal page includes the relevant figures and tables for your reference.

---

### Author Response · Authors · 2024-08-14
**Response Uploaded**

Thank you for all the valuable feedback.

We have uploaded the responses according to the reviewers' requests on the global rebuttal page and individual comments.

Today is the only day left for discussion, so please verify our responses, and if we have addressed your concerns satisfactorily, we kindly request a positive reassessment.

---

### Decision · Program_Chairs · 2024-09-25

**Decision:**

Accept (poster)

**Comment:**

The reviewers generally agree that the manuscript is well written and is clearly motivated, that the proposed method is simple, has novelty, and is reproducible, and that the results and visualizations are compelling. An extensive rebuttal was submitted, which addressed a majority of the reviewers concerns, and also demonstrated that the proposed approach could benefit a different task (temporal action detection). After this phase, R-K2gT raised their rating to borderline-accept and the other three reviewers maintained their ratings. R-FKQN remained negative (borderline reject) but was not active during any of the discussion phases. Therefore, the AC reviewed the paper in detail, finding the review of R-aJYL to be the most compelling. The paper identifies a weakness in existing approaches, proposes a novel solution, and demonstrates a positive performance impact. Further, the AC found the weaknesses identified by R-FKQN to be insufficient reasons for rejection and adequately addressed by the authors. For example, multi-scene pose regression is an established research area and does not need to be re-motivated by the authors.  Additionally, penalizing the proposed approach for only supporting attention-based methods (i.e., transformers) is not viewed as a valid complaint, especially in the current era. As such, the AC reached an accept decision. Please take the reviewer feedback into account when preparing the camera ready version.